# Preclinical Evaluation and Advancements in Vascularized Bone Tissue Engineering

**DOI:** 10.3390/biomimetics10070412

**Published:** 2025-06-20

**Authors:** Toshiyuki Kawai

**Affiliations:** Department of Orthopaedic Surgery, Kyoto University Graduate School of Medicine, 54 Shogoin-kawahara-cho, Sakyo-ku, Kyoto 606-8507, Japan; kawait@kuhp.kyoto-u.ac.jp

**Keywords:** vascularized bone graft, bone tissue engineering, 3D bioprinting, electrospun nanofibers, angiogenesis, osteogenesis, mesenchymal stem cells (MSCs), endothelial cells, growth factor delivery, preclinical animal models

## Abstract

Large segmental bone defects present significant challenges due to the insufficient vascularization of implanted grafts, necessitating advances in vascularized bone tissue engineering. Recent innovations focus primarily on enhancing graft vascularization through advanced biomaterial scaffolds, precise three-dimensional (3D) bioprinting technologies, biochemical interventions, and co-culture techniques. Biomaterial scaffolds featuring microchannels and high-surface-area architectures facilitate endothelial cell infiltration and subsequent vessel formation. Concurrently, sophisticated 3D-bioprinting methods, including inkjet, extrusion, and laser-assisted approaches, enable the precise placement of endothelial and osteogenic cells, promoting anatomically accurate vascular networks. Biochemical strategies that utilize the simultaneous delivery of angiogenic factors (e.g., vascular endothelial growth factor) and osteogenic factors (e.g., bone morphogenetic protein-2) effectively couple angiogenesis and osteogenesis. Additionally, co-culturing mesenchymal stem cells and endothelial progenitors accelerates the development of functional capillary networks. Preclinical studies consistently demonstrate superior outcomes for prevascularized grafts, as evidenced by enhanced vascular inosculation, increased bone formation, and improved mechanical stability compared to non-vascularized controls. These technological advancements collectively represent significant progress toward the clinical translation of engineered vascularized bone grafts capable of addressing complex and previously intractable bone defects.

## 1. Introduction

Bone is a highly vascularized tissue, and its blood supply is crucial for bone development, remodeling, and healing [1]. Consequently, bone regeneration and angiogenesis are tightly coupled processes. Blood vessels deliver essential oxygen, nutrients, and growth factors while providing paracrine signals that guide new bone formation [2]. Large segmental bone injuries that exceed the body’s self-repair capacity (so-called critical-size defects) cannot heal spontaneously and often require grafting or other interventions [3]. A key reason is the difficulty of re-establishing a blood supply at the defect site. Without rapid vascularization, the regenerative process is severely impaired. Insufficient or slow development of a vascular network leads to inadequate nutrient delivery, delaying osteogenesis or even preventing the formation of new bone [2]. In tissue-engineered bone grafts, especially, the absence of early vascular ingrowth can cause central regions to become hypoxic and necrotic, resulting in the death of implanted cells [2]. Indeed, the lack of vascularization has been identified as one of the primary challenges holding back the clinical success of large engineered bone grafts for decades [4]. Overcoming this vascular supply problem is essential for any strategy aimed at regenerating large bone defects.

The current clinical gold standard for repairing sizable bone defects is autologous bone grafting, often using vascularized bone transfers from the patient (e.g., a fibula or iliac crest flap) [5]. Autologous grafts provide living bone tissue with osteogenic cells and growth factors, but they are limited by significant drawbacks. Donor tissue availability is finite, and harvesting bone leads to donor-site morbidity, complications, and pain [6]. Even vascularized autografts require extensive surgery and are constrained in the volume and shape that can be transplanted. Allogeneic bone grafts (from donors) are an alternative that can supply larger volumes, avoiding the issue of limited patient tissue. However, allografts carry risks of immune rejection and disease transmission and typically lack the cellular viability and integrative capacity of autografts [2]. Moreover, neither approach guarantees a pre-existing microvascular network within the graft; both rely on the host to slowly vascularize the transplanted bone, which is especially problematic in large defects. These limitations of autografts and allografts, including limited supply, donor site morbidity, immune complications, and delayed vascular integration, underscore an urgent need for new solutions [4].

Tissue engineering (TE) offers a promising avenue to address large bone defects by creating customized bone grafts that can potentially overcome the shortcomings of traditional grafts [7]. Advanced TE techniques allow for the generation of living bone tissue constructs tailored to a patient’s specific needs [4]. Crucially, for defects beyond a certain size or in environments with poor circulation, engineered bone grafts must include an intrinsic vascular network to ensure the survival of cells and rapid integration after implantation [8]. Recognizing this requirement, numerous strategies have been explored to vascularize engineered bone constructs [4]. Approaches reported in the literature include integrating microchannels or porous networks into scaffolds to serve as templates for blood vessel ingrowth, using methods like 3D printing/bioprinting to create vascular structures within bone grafts, and seeding scaffolds with endothelial or other vessel-forming cells that self-assemble into capillary networks [9]. The incorporation of pro-angiogenic factors (such as vascular endothelial growth factor, VEGF) into bone scaffolds is another widely studied strategy to stimulate rapid neovascularization [10]. In addition, in vivo prevascularization has emerged as a compelling technique. The scaffold or construct is implanted into a well-vascularized environment (for example, a muscle or subcutaneous pocket) prior to use, allowing host blood vessels to penetrate and mature throughout the graft [11]. Once transplanted into the bone defect, such prevascularized grafts have an immediate blood supply through anastomosis, which can greatly enhance graft survival and integration.

Recent advances in vascularized bone tissue engineering have demonstrated the potential of these approaches to overcome critical size defects. Notably, one high-impact study created a large, living bone graft by combining osteogenic and vasculogenic cell populations with growth factors in a scaffold, then prevascularizing it with an arteriovenous loop in vivo [12]. The prevascularized construct was successfully transplanted to heal a critical segmental bone defect in a load-bearing location, serving as a proof-of-concept for a transplantable, patient-specific vascularized bone graft [12]. Such results highlight the state-of-the-art progress in fabricating vascularized bone grafts and underscore the translational promise of engineered grafts that come pre-equipped with functional blood vessel networks. With continuing developments in biomaterials, cell engineering, and fabrication techniques, tissue-engineered vascularized bone grafts are poised to address the longstanding vascularization challenge, offering the prospect of more effective and reliable treatments for large bone defects beyond the limits of autologous grafting.

Table 1 provides an overview of the main 3D-printing technologies (inkjet, extrusion, and laser-assisted), summarizing their benefits and drawbacks for vascularized bone constructs.

In this review, we build on the current body of the literature by providing an in-depth examination of prevascularization strategies in bone tissue engineering. Rather than introducing entirely novel concepts, we synthesize recent advances in engineered vascularized bone constructs and critically discuss how these prevascularization approaches address the enduring challenge of vascularizing large bone grafts. Through this approach, the review aims to serve as a meaningful and integrative contribution to the field by consolidating current insights and highlighting opportunities for future research and clinical translation.

## 2. 3D Bioprinting Approaches

### 2.1. Comparison of Bioprinting Techniques (Inkjet, Extrusion, and Laser-Assisted)

An overview of Inkjet, Extrusion, and Laser-Assisted 3D bioprinting is shown in Figure 1.

#### 2.1.1. Inkjet Bioprinting (Droplet Based)

Inkjet (or droplet-based) bioprinting uses thermal or piezoelectric actuators to eject tiny droplets of cell-laden bioink onto a substrate in a controlled pattern [13]. This approach is fast and cost-effective, with printers readily available, and it can achieve fine placement of cells due to small droplet size (tens of picoliters) [13]. Inkjet printing typically maintains high cell viability (>85%) because it imposes minimal mechanical stress, though some shear or thermal stress can occur during droplet ejection [13]. A key limitation is that bioinks must have low viscosity to form droplets, restricting the use of highly viscous or particulate-laden inks (e.g., those with ceramic powders for bone) [15]. Nozzle clogging is another concern, especially when using bioinks with cells or aggregates [15]. In the context of vascularized bone bioprinting, inkjet is well-suited for the precise deposition of cells or growth factors rather than building a bulk scaffold structure. For example, inkjet bioprinting can precisely pattern endothelial cells to form microvasculature within a bone graft or deposit growth factors in specific regions to induce vascularization [17]. Its high resolution (droplet placement ~50–100 µm) allows for fine control of microarchitectural features, though forming large, load-bearing bone structures solely by droplets can be challenging. Inkjet bioprinting has been successfully used in hybrid strategies, such as printing endothelial cells into an existing bone matrix printed by other means [17], leveraging its precision to prevascularize engineered bone tissues.

Meanwhile, ongoing work has highlighted the importance of innervation in bone repair, suggesting that the inkjet-printed patterns of neurotrophic factors could further improve outcomes in vascular–neural coupling [19].

#### 2.1.2. Extrusion Bioprinting

Extrusion-based bioprinting dispenses a continuous filament of bioink through a nozzle via pneumatic or mechanical (piston/screw) force [20]. It is the most widely employed technique for tissue constructs, including bone, because of its ability to print with a wide range of bioink viscosities and compositions [21,22]. High-viscosity hydrogels, polymer pastes, or composites (e.g., bioinks with β-tricalcium phosphate or hydroxyapatite for bone) can be printed, allowing for the creation of robust, load-bearing structures [23]. Extrusion can also deposit high densities of cells, even spheroids or tissue strands, directly into the construct [23]. A major advantage for vascularized bone is that extrusion can create large, anatomically shaped scaffolds with embedded channels or heterogeneous regions; techniques like coaxial extrusion enable the direct printing of perfusable vascular tubes within bone constructs [24]. However, extrusion bioprinting generally has lower resolution (feature size typically > 100 µm) compared to inkjet or laser methods, since nozzle diameters and filament spreading limit the minimum structure size [14]. The cell viability can also be lower. The shear forces and pressure during extrusion can reduce viability to ~40–86%, especially with small nozzle diameters or high speeds [14]. Despite these limitations, extrusion’s ability to fabricate multi-material, mechanically robust bone constructs with interwoven vasculature makes it indispensable. Indeed, extrusion is often used to print the bulk of a bone tissue (e.g., a hydrogel matrix with osteogenic cells), sometimes in tandem with sacrificial bioinks to leave vascular channels that are later endothelialized [16]. Its compatibility with both natural bioinks and thermoplastics (for scaffold support) allows for combining a cell-laden phase for biology and a polymer phase for mechanical strength into one print, an approach frequently applied in bioprinted bone scaffolds [18].

In addition, new work using nanosilicate-functionalized polycaprolactone scaffolds in extrusion-based printing has shown accelerated vascularized bone regeneration in segmental defects, illustrating the synergy between advanced biomaterials and extrusion [25].

#### 2.1.3. Laser-Assisted Bioprinting (LAB)

Laser-assisted bioprinting uses a pulsed laser to propel droplets of bioink from a donor slide to a receiving surface in a nozzle-free manner [26]. The laser pulse creates a high-pressure vapor bubble that ejects a cell-laden ink droplet with micron-scale precision [26]. LAB offers the highest resolution of the three techniques. Droplet placement can be on the order of tens of micrometers or even single-cell resolution, since the process is limited only by laser focal spot size and energy control [27]. This ultra-high resolution enables printing very fine vascular patterns or cell arrangements that mimic capillaries. Another key advantage is excellent cell viability. Because it is a contact-free method (no nozzle), there is minimal mechanical stress on cells; studies report viability often above 90–95% post-printing [27]. LAB can also handle higher viscosity bioinks than inkjet (no clogging concerns), including viscous collagen or nano-hydroxyapatite gels relevant for bone, as long as they can form a thin film on the donor slide [27]. The main drawbacks are its complexity and scalability. The setup (known as a “ribbon” with an absorbing layer) requires careful preparation and alignment [28], and printing large constructs can be time-consuming since each droplet is ejected individually. Thus, LAB is often used to pattern cells or small-volume features rather than to print an entire bulk scaffold. In vascularized bone engineering, LAB is particularly useful for in situ cell patterning, such as, for example, printing osteoprogenitor cells and endothelial cells in defined microscale arrangements within a defect site to encourage organized bone and vessel formation [29]. LAB’s precision has enabled studies where mesenchymal stem cells were laser-printed with microscale accuracy into bone defects, resulting in enhanced and spatially organized bone regeneration [29]. While not as rapid for producing cm-scale grafts, laser bioprinting provides unmatched control at the cell and capillary level, complementing extrusion techniques. Researchers are exploring hybrid systems that use extrusion for bulk bone structure and laser printing to seed endothelial cells in fine vascular networks, marrying the strengths of both approaches [18,30].

Furthermore, LAB has recently been leveraged to deposit both osteogenic cells and angiogenic signals into large bone defects, facilitating in vivo vascular anastomosis more rapidly [3].

Overall, each bioprinting modality offers distinct advantages for vascularized bone constructs. Inkjet excels in speed and cell placement precision but is limited by material viscosity. Extrusion prints durable, large structures with diverse bioinks, but with lower resolution, and laser-assisted printing achieves the highest resolution and viability, ideal for microvascular features, albeit with higher complexity [31,32]. Often, these methods are used in combination to capitalize on their complementary strengths in creating vascularized bone tissue. For instance, a bone scaffold might be extruded from a cell-laden mineralized hydrogel. Then, an inkjet or laser printer adds endothelial cells or growth factors into channels or specific regions to establish a vascular network. This strategic coupling has been highlighted as a way to overcome the challenge of integrating vasculature into thick bone grafts [17]. Table 2 summarizes the key cell types commonly employed in vascularized bone tissue engineering, including their roles in osteogenesis or angiogenesis, and the relevant literature references.

### 2.2. Preclinical and In Vivo Successes in Vascularized Bone Bioprinting

A number of preclinical studies have demonstrated the feasibility and benefits of 3D-bioprinted vascularized bone grafts in vivo. These studies typically combine osteogenic cells (or progenitors) with endothelial cells to prevascularize the construct, reporting improved outcomes in terms of vascular infiltration and bone regeneration.

**Kériquel et al. (2017)** applied laser-assisted bioprinting in situ to treat a critical-size bone defect, illustrating the precision of LAB for prevascularization [33]. In this study, mesenchymal stromal cells (bone progenitors) were laser-printed with a collagen/nanohydroxyapatite bioink directly into a mouse calvarial defect with different spatial patterns [33]. The in vivo results showed that the printed cells survived well, and the spatial arrangement of the bioprinted cells influenced new bone formation, suggesting that a precise placement of cells can guide the healing process [33]. Notably, the LAB-enabled constructs led to faster and more organized bone regeneration than non-patterned cell delivery, and histological analysis confirmed bone tissue formation in alignment with the printed geometry. In a follow-up study, the same group laser-printed endothelial and bone progenitor cells together (creating an in situ prevascularized implant), which led to improved vascularization and bone healing compared to implants without prevascularization [34]. These findings underscore LAB’s potential in bone defect surgery. A surgeon could directly print a tailored pattern of osteogenic and vasculogenic cells into a defect, achieving rapid vascularized bone repair. This study provided the first demonstration of in situ laser-assisted bioprinting of cells directly into a bone defect, showing that precise MSC deposition (and even the printed cell pattern geometry) can significantly influence bone regeneration outcomes. However, the approach was only validated in a small, planar calvarial mouse defect, and the LAB technique remained constrained to flat surfaces, highlighting the need for further technological advances to enable integration of this strategy into large or geometrically complex bone repairs;**Rukavina et al. (2020)** bioprinted prevascularized bone patches by co-printing human adipose-derived stem cells (hASCs) in an osteogenic hydrogel (via extrusion) alongside human umbilical vein endothelial cells (HUVECs) deposited in defined locations (via drop-on-demand inkjet) [35]. The cuboid constructs were implanted subcutaneously in mice. After implantation, the pre-seeded HUVECs formed human-derived microvessels of various diameters within the graft, which connected with the host vasculature (evidenced by mouse pericytes lining the human vessels) [35]. Simultaneously, the ASC-laden portion of the graft produced a calcified bone matrix, indicating new bone tissue formation [35]. This study demonstrated that bioprinting a bone construct with prevascularization (i.e., both vessel and bone cells patterned together) leads to functional microvessels and a mineralized matrix in vivo, a promising step toward treating critical-size bone defects with engineered grafts. This work demonstrated a bioprinting strategy for prevascularized bone constructs by co-printing mesenchymal stem cells with endothelial cells, resulting in the formation of human-derived microvessels (stabilized by host pericytes) throughout the construct, along with a calcified bone matrix deposited by the MSCs upon implantation. A key limitation is that the proof-of-concept was shown in an ectopic subcutaneous mouse model (yielding ectopic bone formation) rather than an orthotopic bone defect, so its efficacy in true load-bearing bone repair and full vascular integration within a physiological environment remains unproven;**Kim et al. (2022)** fabricated a multicellular construct for spinal fusion by extrusion-printing a bioink composed of collagen + β-tricalcium phosphate (osteoconductive ceramic) laden with hASCs and HUVECs [36,37]. The 3D-bioprinted scaffold had a lattice of microscale struts, effectively creating a porous bone substitute with evenly distributed stem and endothelial cells. In vitro, the co-culture construct showed synergistic crosstalk. The presence of HUVECs induced a strong angiogenic phenotype (e.g., upregulation of VEGF and CD31), while the stem cells exhibited enhanced expression of osteogenic markers, indicating concurrent vascular and bone differentiation [36]. When implanted in a mouse spinal fusion model, the prevascularized bioprinted graft achieved significantly greater new bone formation and higher vessel density at the fusion site compared to printed constructs with stem cells alone [36,37]. The inclusion of endothelial cells accelerated host perfusion into the scaffold and supported the maturing bone tissue, resulting in improved spinal fusion outcomes. This study developed a novel bioprinted hybrid construct that integrated endothelial cell spheroids into a human stem cell-laden scaffold, which achieved markedly enhanced vascular network formation and new bone deposition in vivo compared to a conventional cell-mixed bioprinted graft. Nonetheless, the approach requires labor-intensive spheroid fabrication and multi-step assembly, and it was tested only in a specialized rat mastoid defect model. These factors may limit its immediate translational applicability to large, critical-sized bone defects;**Goker et al. (2024)** demonstrated a single-step bioprinting strategy for vascularized bone using a growth factor delivery approach in vitro [38,39]. They printed bone constructs with human ASCs in a dual-factor-laden bioink, with microspheres of BMP-2 in regions designated to become bone and VEGF in regions designated to become vessels [38]. This spatially controlled release of inductive factors caused the ASCs to differentiate into osteoblasts in the BMP-2 zones (expressing osteopontin) and into endothelial-like cells in the VEGF zones (expressing CD31) [38]. The result was a patterned construct where mineralized bone-like tissue and vessel-like structures developed side by side from a single cell source, verified after culturing the construct in a perfusion bioreactor [38,39]. This recent study introduced a spatially controlled growth-factor delivery approach in a 3D-bioprinted bone graft, using microencapsulated BMP-2 and VEGF within a single ASC-laden bioink to induce localized osteogenic and vasculogenic differentiation of the same stem cell population. The main limitation is that the concept was validated only in perfused in vitro cultures (bioreactor), without any in vivo implantation, so the ability of these single-cell-source constructs to heal actual bone defects and anastomose with host vasculature remains to be demonstrated.

Collectively, these preclinical successes highlight that 3D-bioprinted vascularized bone grafts can support blood vessel formation in vivo, significantly increase bone repair in critical defects, and establish organized vascular networks that integrate with host circulation. The convergence of vascularization and osteogenesis, whether through co-printing multiple cell types or patterning cells and growth factors, consistently yields better outcomes than bone constructs without prevascularization. Preclinical studies in small animal models lay a strong foundation for future translation. Ongoing research is extending these approaches to larger animal models and refining the maturation of printed vascular networks (for instance, by perfusing bioprinted grafts in bioreactors before implantation) to ensure that 3D-bioprinted vascularized bone grafts can effectively heal large, clinically relevant bone defects [15,36]. The integration of advanced bioprinting techniques with clever bioengineering (such as sacrificial vascular channels, coaxial printing of vessel tubes, or growth factor patterning) continues to improve vascularized bone constructs, moving this technology closer to clinical application in regenerative orthopedics.

## 3. Electrospining

Electrospun nanofiber scaffolds merit separate consideration due to their unique fabrication advantages, including the ability to closely mimic the extracellular matrix (ECM) architecture and maintain the integrity and functionality of sensitive bioactive molecules such as growth factors, thereby significantly enhancing vascularization potential.

Electrospun fibers have a high surface area and porosity, and their architecture can be tuned (e.g., fiber diameter, alignment) to influence cell behavior [40,41]. The simplicity and versatility of electrospinning have made it a popular method for bone TE scaffolds [42]. However, a typical electrospun mat has tightly packed fibers with small pore sizes, which can limit cell infiltration and vascularization. Unmodified nanofiber scaffolds do not inherently favor endothelial cell attachment, making it challenging to form blood vessels unless further modified [43,44]. Therefore, researchers have focused on functionalizing electrospun scaffolds to enhance vascularized bone regeneration.

One strategy is to incorporate micro- and macro-porosity into electrospun constructs. For example, creating a reticular (net-like) fiber structure or multilayered fiber meshes can facilitate host vessel invasion. Jin et al. found that electrospun membranes with a more open, reticular architecture promoted angiogenesis, macrophage recruitment, and osteogenesis, compared to densely packed fibers [45]. Another approach is combining electrospun fibers with other scaffold elements (e.g., electrospinning onto a 3D-printed frame) to introduce larger channels for vessel ingrowth. Additionally, advanced electrospinning techniques like coaxial electrospinning allow for encapsulating bioactive agents in core–shell nanofibers for sustained release of growth factors. For instance, core–shell fibers have been loaded with dual factors (e.g., an angiogenic factor in one layer and an osteogenic factor in another) to achieve a sequential delivery that supports both vessel formation and bone formation over time [46,47]. Such dual-drug electrospun mats have shown superior vascularized bone regeneration in comparison to single-factor controls in preclinical studies.

The chemical functionalization of nanofibers can also drastically improve vascularization. Neves et al. demonstrated a notable example by covalently immobilizing growth factors on an electrospun PCL membrane [48]. They used carbodiimide chemistry to attach anti-VEGF and anti-BMP-2 antibodies onto PCL nanofibers, then bound VEGF and BMP-2 to these sites [40,48]. MSCs cultured on this dual-functionalized scaffold showed elevated alkaline phosphatase (ALP) and endothelial gene expression, indicating simultaneous osteogenic and vasculogenic differentiation [40]. In a chick chorioallantoic membrane (CAM) assay, scaffolds with both VEGF and BMP-2 induced a dense, mature vascular network surrounding the implant, far more extensive than scaffolds with either factor alone [40,48]. This illustrates the potency of combining angiogenic and osteogenic cues on electrospun fibers to create a pro-vascularization microenvironment. Notably, electrospun fibers can be made from various materials (natural polymers like collagen, silk fibroin, chitosan, or synthetic polymers like PCL and PLGA), and composite fibers (e.g., polymer–mineral blends) further enhance osteointegration [49,50]. For example, coating nanofibers with bioactive hydroxyapatite or silica can improve their osteoconductivity [51], while also providing a mineral phase that may favor endothelial cell adhesion. Overall, electrospinning offers a versatile platform for engineering bone scaffolds, but often requires additional architectural modifications or biochemical functionalization to achieve efficient vascularization [52,53]. Table 3 outlines the main growth factors highlighted in this study, e.g., VEGF, BMP-2, and FGF-2, along with their core functions in coupling angiogenesis and osteogenesis.

## 4. Conventional Scaffold-Based Strategies

Beyond bioprinting and electrospinning, a variety of scaffold fabrication techniques have been explored to promote vascularized bone growth. These include traditional porous scaffolds (e.g., foams, sponges, decellularized bone matrices, and bioactive ceramics), as well as newer microfabrication approaches. A key principle in scaffold design is to provide an appropriate 3D architecture that facilitates blood vessel invasion. Highly porous scaffolds with interconnected macropores (>100 µm) are known to support quicker vascular ingrowth and bone formation. For example, a 3D-printed “sponge” scaffold with layered, interconnected pores significantly enhanced cell ingrowth and paracrine signaling from MSCs, leading to well-vascularized bone regeneration in vivo [54]. Similarly, gradient porosity designs—having larger pores on the periphery of an implant—have been shown to promote angiogenesis at the implant interface, improving overall osteogenesis compared to uniformly small-pore designs [55]. These findings underscore that scaffold microstructure (pore size, geometry, and interconnectivity) is a critical determinant of vascularization efficiency.

Researchers have also integrated micro-channel networks into bone scaffolds using techniques like gas foaming, sugar particle leaching, or direct 3D plotting of sacrificial filaments. Such channels act as conduits for host vessels to penetrate deep into the construct. In one study, preformed microvascular networks in a scaffold were able to rapidly anastomose with the host vasculature after implantation, establishing blood flow through the graft [56]. This rapid inosculation is vital to avoid the long lag time of spontaneous vessel ingrowth. Another strategy is the in vivo pre-vascularization of scaffolds: for instance, implanting a cell-seeded scaffold into a well-vascularized tissue (like omentum or muscle) or creating an arteriovenous loop inside a chamber with the scaffold, to induce vessel ingrowth before transferring the construct to the bone defect. Such in vivo bioreactor approaches have successfully generated vascularized bone grafts that later integrate with the surrounding bone [57]. While not a “fabrication” technique per se, this strategy relies on the body’s own angiogenic response to fabricate a vascular network within the graft prior to final implantation.

In terms of materials, conventional bone scaffolds have spanned a wide range, from natural polymers (collagen, gelatin, fibrin) and ceramics (hydroxyapatite, tricalcium phosphate) to synthetic polymers (PLGA, PEG, polyurethane) and composites. The choice of material influences not only mechanical properties and biodegradation but also vascularization. Bioactive materials can stimulate angiogenesis. For example, incorporating angiogenic trace elements like cobalt or copper into ceramic scaffolds triggers a hypoxia-like response that upregulates VEGF, thereby promoting neovascularization [4]. Likewise, surface modification of a scaffold (e.g., with heparin or ECM-derived peptides) can enhance endothelial cell attachment and growth factor binding, aiding vessel formation. Researchers have combined multiple modifications, including physical (architectural), chemical (factor delivery), and biological (cell seeding), to maximize vascularization [58]. The consensus from two decades of studies is that no single component is sufficient. Instead, the most successful scaffold-based approaches integrate optimized materials, pore structure, and bioactive signals to create an environment conducive to both blood vessel and new bone formation [2,59].

## 5. Influence of Growth Factors on Vascularization

Growth factors are pivotal signaling molecules for orchestrating angiogenesis and osteogenesis in bone repair. Among the myriad factors, vascular endothelial growth factor (VEGF) and bone morphogenetic protein-2 (BMP-2) have been the most extensively studied in the context of vascularized bone grafts [60]. VEGF is a potent angiogenic factor that specifically acts on endothelial cells to stimulate blood vessel formation, whereas BMP-2 strongly induces an osteogenic differentiation of progenitor cells [10,60]. The interplay of these (and other) growth factors in normal bone healing is well known. Osteoblasts secrete VEGF to recruit blood vessels, and endothelial cells, in turn, produce BMPs and other factors that promote osteoblast maturation [61]. This natural coupling suggests that delivering both angiogenic and osteogenic factors could synergistically enhance engineered bone graft outcomes.

### 5.1. VEGF Delivery

Simply adding VEGF protein into a scaffold or defect (e.g., bolus injection) often fails to produce lasting vasculature due to its short half-life and the need for sustained gradients [61]. Early studies found that bolus VEGF could induce aberrant, disorganized vessels that regressed quickly [61]. To address this, researchers developed controlled delivery systems, encapsulating VEGF in microspheres, hydrogels, or coatings, to provide prolonged release. For example, Formiga et al. showed that incorporating VEGF into degradable PLGA microspheres yielded stable, long-lasting blood vessels in vivo, unlike the transient vasculature from free VEGF injection [62]. Controlled VEGF release ensures a persistent angiogenic stimulus, allowing nascent vessels to mature and integrate with the host circulation [62].

### 5.2. BMP-2 Delivery

BMP-2 is widely used to stimulate bone formation, but its high dose requirements and potential side effects (e.g., ectopic bone) necessitate careful delivery. Sustained BMP-2 release from scaffolds (via collagen sponges, microspheres, or layer-by-layer coatings) has been shown to improve local bone regeneration by maintaining osteogenic signaling [63,64]. Interestingly, BMP-2 can also indirectly aid vascularization. It upregulates angiogenic factors in osteogenic cells and can attract endothelial progenitors [61,64].

### 5.3. Combined and Sequential Delivery

Given the interdependence of angiogenesis and osteogenesis, several studies have investigated the co-delivery of VEGF and BMP-2 (or other factor combinations). Dual delivery often yields synergistic effects on vascularized bone formation. Richardson et al. reported that delivering VEGF and PDGF together produced a much higher density of stable microvessels than delivering either alone [65]. Patel et al. encapsulated VEGF and BMP-2 in injectable gelatin microparticles and observed a synergistic enhancement of bone regeneration. Early bone formation was significantly greater with both factors than with BMP-2 alone [66]. Similarly, a scaffold with a sequential release coating (VEGF released in the first week, BMP-2 over two weeks) led to a ~33% increase in bone mineral density compared to BMP-2 alone in vivo [67]. Notably, Mikos and coworkers found that in a rat calvarial defect, early (4-week) bone fill was higher with dual VEGF+BMP-2 delivery versus BMP-2 alone, although by 12 weeks, the differences leveled off [68]. This suggests that VEGF accelerates the initial phase of bone repair by establishing a blood supply, which in turn supports BMP-2-driven osteogenesis; over longer periods, even BMP-2 alone eventually benefits from host angiogenesis that develops, but the healing timeline is shortened with the angiogenic factor [68]. Beyond VEGF and BMP-2, other growth factors have been explored. Fibroblast growth factor-2 (FGF-2) can promote both endothelial and osteoprogenitor activity [69,70], platelet-derived growth factor (PDGF) recruits progenitors and perivascular support cells, and stromal-derived factor-1 (SDF-1) helps mobilize vascular progenitors. Ultimately, tailoring the spatiotemporal delivery of multiple growth factors in scaffolds, often through sophisticated carriers or bioprinting of factor gradients, has proven to be an effective strategy to maximize vascularization efficiency in engineered bone grafts.

## 6. Cell Sources and Cellular Interactions

The cellular component of tissue-engineered bone is as important as the scaffold and growth factors. To create pre-vascularized bone constructs, two main cell types are typically combined: vascular endothelial cells (or their progenitors) to form blood vessels and osteogenic cells (such as osteoblasts or mesenchymal stem cells) to form bone tissue [71]. The dynamic crosstalk between these cells drives the simultaneous development of vasculature and bone matrix. In a co-culture, osteogenic cells secrete pro-angiogenic factors like VEGF that stimulate endothelial cells, while endothelial cells can secrete osteogenic factors (e.g., BMPs) and mitogens that enhance osteoblast differentiation [72,73]. This bidirectional signaling leads to emergent behaviors, for instance, endothelial cells aligning into capillary-like structures alongside regions of mineralized matrix deposition.

Mesenchymal stem cells (MSCs) are widely used as the osteogenic component due to their capacity for osteoblastic differentiation and ease of isolation (from bone marrow or adipose tissue) [74]. MSCs also release trophic factors that support angiogenesis. Endothelial cells used in bone TE are often human umbilical vein endothelial cells (HUVECs) or microvascular endothelial cells; these can form tube networks in vitro when cultured in 3D environments. Endothelial progenitor cells (EPCs) or endothelial cells derived from stem cells have also been tested. Some strategies include co-culturing MSCs and ECs on scaffolds in ratios optimized for vessel formation. A common approach is a 1:1 or 2:1 ratio, which has been shown to produce robust capillary networks in vitro [75,76]. Encapsulating both cell types in a hydrogel can yield a spontaneous self-assembly of vessel-like structures within days, especially under proper growth factor stimulation or hypoxic preconditioning [77,78].

Cellular interactions are not limited to chemical signals. Physical contact via direct cell–cell junctions also plays a role in co-cultures. Gap junction communication between osteoblasts and endothelial cells has been found to regulate gene expression and promote osteogenic differentiation [79,80]. For example, connexin-mediated coupling can synchronize the activity of adjacent bone and vessel cells, mimicking the in vivo niche conditions. Moreover, the presence of a developing vasculature can modulate the behavior of other cells in the construct, such as pericytes or smooth muscle cells (if included), which wrap nascent endothelial tubes and stabilize them. Recent studies have even identified specialized subpopulations, like Type H endothelium in bone, which have high ALP expression and directly stimulate osteoprogenitors. Recreating such a specific endothelium in engineered grafts may further improve outcomes [81,82].

In practice, successful prevascularization often involves tri-culture or supporting cell types, e.g., adding fibroblasts or stromal vascular fraction cells to provide angiogenic growth factors or using osteoclast precursors to remodel the matrix and signal to vessels. Tissue spheroids (microtissues) composed of MSCs and ECs have been bioprinted to expedite network formation, as spheroids bring cells into immediate 3D contact at high density [83,84]. Regardless of the method, the evidence is clear that co-culturing vascular and bone cells yields constructs that vascularize more readily and produce more bone in vivo than monocultures [85,86]. For example, one study showed that adding endothelial cells to self-assembled human bone-like tissue constructs led to a dense capillary network in vitro within 2 weeks, and when implanted in rats, the prevascularized tissue had improved cell survival and bone repair compared to constructs without endothelial cells [87]. Thus, leveraging the synergistic interactions between cell types is a cornerstone of engineering vascularized bone grafts.

## 7. Preclinical Models and Performance Outcomes

Preclinical in vivo studies, mainly in animals, have been indispensable for evaluating the performance of vascularized bone grafts. Small animal models such as rodents (mice and rats) are most commonly used to test vascularization efficiency and the osteointegration of engineered constructs. Typical models include rat calvarial (skull) critical-size defects, rat femoral segmental defects, or mouse subcutaneous implantation for ectopic bone formation. These models allow for analysis of neovascularization within the graft and the extent of new bone bridging over weeks to months. Vascularization efficiency is assessed by methods like micro-computed tomography (microCT) with contrast agents to visualize blood vessels, histological staining of endothelial markers (CD31, vWF), and quantification of vessel density or perfusion in the graft. Osteointegration is evaluated by new bone formation (via microCT or histomorphometry), mechanical testing, and the continuity between native and grafted bone.

Many studies report that pre-vascularized bone implants lead to superior vascularization and bone healing in vivo compared to non-vascularized controls. For instance, Cunniffe et al. showed that a bioprinted MSC-laden construct containing genes for angiogenic factors, when implanted subcutaneously in rats, led to significantly greater vascularization and mineralization than acellular implants [88]. In a rat cranial defect model, an electrospun nanofiber scaffold functionalized with osteogenic coatings achieved ~94% new bone fill at 6 weeks, versus much lower bone formation in unmodified scaffolds [89]. This dramatic improvement was attributed to the enhanced vascular invasion and osteoconductivity of the modified scaffold. Prevascularization also improves the speed and quality of integration. As noted earlier, bioprinted grafts with preformed capillaries can inosculate with host vessels within a few days [81], supplying oxygen and nutrients that keep the core of the graft alive. Nulty et al. observed that prevascularized implants not only formed more blood vessels in a critical femoral defect, but those vessels supported greater new bone volume, effectively bridging a defect that would otherwise not heal [90,91].

Researchers have explored larger animal models (e.g., rabbit radius defects, sheep segmental defects) to move closer to clinical translation, though such studies are fewer. Nonetheless, the principles from rodent models generally hold. Grafts that include vascularization strategies (cells or factors) consistently show better osteointegration. Improved metrics include higher bone–implant contact percentages, earlier mineralized matrix deposition, and stronger mechanical properties of the repaired bone. For example, in an immunocompromised mouse model of calvarial repair, laser-bioprinted patches containing MSCs, ECs, and VEGF led to significantly more bone formation and vessel density at 8 weeks than patches without ECs [92]. In a rat femoral defect, a prevascularized 3D-printed hydrogel implant supported blood flow restoration across the defect and yielded a greater bone union rate than a non-prevascularized implant [15]. Such outcomes underscore that vascularization efficiency strongly correlates with osteogenesis in bone tissue engineering. Constructs that rapidly establish a perfused network avoid the central necrosis that plagues large implants. Thus, all regions of the graft participate in bone regeneration. Over time, the transplanted vascularized graft not only survives but actively remodels and integrates. Host blood vessels infiltrate and connect, while host osteoprogenitors migrate in along those vessels (angiogenesis and osteogenesis are coupled processes). Therefore, the key performance metrics of engineered bone grafts—vascularization and osteointegration—are intimately linked. Improved vascularization is often a prerequisite to robust, long-term osteointegration [91,93].

## 8. Interim Summary

In summary, two decades of research have converged on multi-faceted approaches to engineer vascularized bone grafts. Advances in fabrication techniques have enabled more biomimetic constructs. High-precision bioprinting allows the pre-design of vessel networks and heterogeneous cell distributions, whereas electrospinning and scaffold engineering provide optimized environments for cell attachment and factor delivery. Incorporating angiogenic and osteogenic growth factors (especially VEGF and BMP-2) in a controlled manner has proven crucial for boosting vascularization efficiency without compromising the maturation of bone tissue. Equally important is the inclusion of appropriate cell populations—co-cultures of endothelial and osteogenic cells that communicate and co-differentiate—to kickstart vascular network formation and bone matrix deposition in vitro, thereby priming the graft for successful in vivo integration. Preclinical animal studies overwhelmingly support the premise that prevascularized bone constructs lead to faster and more complete bone healing, with higher vessel densities and better graft–host union than non-vascularized counterparts.

Looking ahead, the field is moving toward a biomimetic integration of these strategies, such as, for example, the biofabrication of hierarchical vascular trees within bone substitutes, gene-activated bioinks that release cues in situ, or smart scaffolds that respond to physiological signals to modulate angiogenesis. Different scaffold materials, from bioceramics to decellularized matrices, are being combined with microfluidic designs and cell sheet engineering to further improve vascularization outcomes. Meanwhile, larger animal studies and early clinical case reports are beginning to validate the translational potential of vascularized tissue-engineered bone grafts. Achieving efficient vascularization is the cornerstone for the next generation of grafts that can be safely scaled to clinical-size defects and load-bearing applications. With continued interdisciplinary efforts in biomaterials, cell biology, and biofabrication, truly biomimetic vascularized bone grafts—ones that closely recapitulate the complexity of native bone’s vasculature and tissue interfaces—are on the horizon [2,94]. Such grafts hold promise to significantly improve patient outcomes in bone regeneration by ensuring that the engineered tissue survives, integrates, and functions as living bone.

## 9. In Vitro Prevascularization for Bone Tissue Engineering: Challenges and Future Strategies

Vascularization is a critical bottleneck in bone tissue engineering (BTE). Preforming vascular networks in vitro (prevascularization) before implantation can improve graft integration by enabling rapid anastomosis with the host circulation [95,96]. However, there are significant challenges to achieving stable, functional vasculature in engineered bone constructs, both in vitro and after in vivo implantation. This section reviews key challenges over the past two decades and discusses emerging strategies in bioreactors, biomaterials, and biochemical cues to enhance prevascularization.

### 9.1. Stability and Maturity of In Vitro Vascular Networks

Forming a robust microvascular network within a bone graft in vitro remains challenging. Engineered capillary networks often lack full structural and functional maturation when grown in vitro. For example, simple co-cultures of endothelial cells (ECs) with osteogenic cells can produce capillary-like structures and boost angiogenic and osteogenic gene expression [54]. Yet, these primitive vessels tend to be limited to two-dimensional cultures and are non-perfusable [97]. Without blood flow or proper supporting cells, the nascent vessels can collapse or regress before implantation. Ensuring lumen formation, basement membrane deposition, and pericyte/smooth muscle cell coverage in vitro is difficult without the physiological cues present in vivo. Researchers have found that an extended static culture in 3D scaffolds can promote more extensive capillary networks [81], but maintaining those networks in a viable state is time-sensitive and labor-intensive [98]. In summary, in vitro prevascularization techniques must strike a balance by providing enough time for vessels to form and stabilize but not being so long that they deteriorate or lose functionality prior to transplantation.

A related hurdle is achieving networks of sufficient density and size to support large bone grafts. Engineered tissues thicker than a few hundred microns develop necrotic cores unless a convective oxygen supply is in place [99,100]. Ideally, a prevascularized construct would have a capillary spacing <200 µm (or even on the order of 20 µm in highly metabolic tissue) so that all cells lie within diffusion distance of a vessel [101]. Reaching this capillary density in vitro is non-trivial. Most studies to date report success in prevascularizing thin or small (<1 mm) constructs, whereas scaling up to clinically relevant volumes remains difficult [100]. The lack of physiological blood pressure and shear stress in the static culture means that the vessels do not remodel into a hierarchical network, and their lumens may remain narrow. In vivo, blood perfusion stabilizes vessels, expands lumens, and prunes the network into an efficient hierarchy; mimicking these processes in vitro is an ongoing challenge [2]. Current in vitro vessels are often immature (leaky, unorganized) and prone to regress without proper support, highlighting the need for improved culture conditions to attain stable, mature microvasculature before implantation [102].

### 9.2. Maintaining Vascular Function After Implantation

Even if a prevascularized construct is achieved in vitro, maintaining its vascular function in vivo poses additional challenges. Upon implantation, the preformed vessels must inosculate (anastomose) with the host’s blood vessels to restore perfusion [100,103]. If this connection is delayed or incomplete, portions of the graft can become ischemic. Rapid graft revascularization is critical for cell survival, yet in practice, it is often not observed to the degree expected [102]. One issue is that the mechanisms of engineered-to-host vessel anastomosis are not fully understood and can be unpredictable [54]. The host may need to actively sprout new capillaries into the graft to connect with the preformed network, a process that still takes time (days) and can be impeded by inflammation or a lack of signaling cues.

Another challenge is ensuring that the transplanted vessels carry blood flow long-term. Even if some connections form, perfusion may be limited to only part of the network, with other segments collapsing if they fail to find a host counterpart. Perfusion limitations and vessel regression can occur if the blood supply through the graft is insufficient or uneven [54,96]. Moreover, without proper arterial supply and venous drainage, the preformed vessels might thrombose or become non-functional. Studies have shown that organizing the engineered vasculature can improve integration. For instance, aligned endothelial cords guided host capillaries to perfuse the graft within ~3 days in vivo [104]. These neovessels progressively matured over 4 weeks with host-derived pericytes, demonstrating that rapid anastomosis is achievable with an optimal prevascular architecture [104]. Nonetheless, achieving seamless integration with the host vasculature in every case is difficult. Strategies like delivering pro-angiogenic factors or using chemoattractants (e.g., SDF-1) have been attempted to encourage host vessel ingrowth [105], as have surgical techniques to directly connect an artery/vein to the construct (in large animal models). While these approaches help, ensuring the long-term perfusion and stability of the graft’s vasculature remains a major hurdle. Indeed, one study using preformed networks from iPSC-derived ECs found that the implanted human vessels did survive and connect with the host (increased vessel density at 8 weeks), but this did not dramatically improve bone formation without additional osteogenic stimuli [100,106]. This suggests that vascular function alone is not sufficient. The vessels must integrate both structurally and functionally into the host’s circulatory system and work in concert with bone regeneration processes.

### 9.3. Cell Survival and Mechanical Compatibility

Prevascularization is ultimately a means to support cell survival in large bone grafts, yet if the transplanted construct’s environment is hostile, cells can still succumb to hypoxia and nutrient deprivation. The initial period post-implantation is critical. Even a prevascularized graft may experience some lag before full perfusion is established. During this window, cells in the construct rely on limited diffusion from the host bed. If the preformed vessels are not flowing blood immediately, the core of the graft can become hypoxic, which is especially relevant in load-bearing bone defects with poor local blood supply. Adequate oxygen and nutrient delivery to all regions is essential [4]. Researchers have explored supplementing constructs with oxygen-releasing materials or using hyperoxic culture to load oxygen prior to implantation [107], but these are short-term fixes. The coupling of angiogenesis and osteogenesis is another consideration. Bone-forming cells (osteoblasts) depend on blood-borne signals and oxygen, while vessels, in turn, rely on signals from osteoprogenitors [81,108]. If either aspect falters (e.g., poor vessel function or insufficient osteogenic activity), healing is compromised.

The mechanical environment of the defect site further complicates matters. Engineered bone constructs often have mechanical properties that are different from native bone, leading to a mismatch in stiffness or stability. This mismatch can affect vascularization in several ways. If a scaffold is significantly stiffer than the surrounding bone, it can shield the tissue from mechanical loads (stress shielding), which may reduce the mechanical stimuli that normally promote bone remodeling and blood vessel development [109]. Osteocytes and bone-lining cells sense mechanical strain. If an implant is too rigid, these cells may receive insufficient stimulus and, thus, produce fewer osteogenic and angiogenic signals [109]. Conversely, if a scaffold is too compliant or lacks strength, it may deform under loads, potentially collapsing nascent vessels or creating micro-motions at the graft–host interface that disrupt vascular connections. In load-bearing sites, inadequate mechanical support can lead to fibrous tissue formation instead of vascularized bone union [109]. Achieving mechanical compatibility means designing scaffolds with elastic moduli and degradation profiles that maintain structural integrity while allowing constructive remodeling. Additionally, the mechanical forces of the in vivo environment (such as pulsatile blood pressure or tissue strain during movement) can influence the implanted vasculature. Native blood vessels adapt their structure (diameter, wall thickness) in response to shear stress and cyclic strain [110]. An engineered vessel network not conditioned to these forces might be prone to hyperpermeability or failure when exposed to them. Overall, the challenge is to create a prevascularized construct that is not only biologically compatible but also mechanically harmonious with the host tissue, providing enough stiffness to support regeneration but not so much as to hinder biophysical signaling or integration. Balancing these factors is critical to maintain cell viability and function after implantation.

## 10. Future Strategies

To surmount the above challenges, researchers are developing advanced strategies that enhance vascular network formation and function in bone grafts. In particular, the use of bioreactors and microfluidic devices can better mature networks in vitro, novel biomaterials and scaffold designs can encourage vessel ingrowth and stability, and smart integration of growth factors and appropriate cell sources can synergistically promote robust vascularized bone formation. These approaches are often combined to recreate a more physiological environment for prevascularization.

### 10.1. Use of Bioreactors and Microfluidic Devices

One promising avenue is deploying specialized culture systems, such as perfusion bioreactors and organ-on-a-chip microfluidic devices, to cultivate vascularized bone tissue under dynamic, physiologically relevant conditions. Unlike static culture, perfusion systems actively pump culture medium through or around the developing tissue, supplying nutrients and oxygen and imparting fluid shear stress on the endothelial cells. This mimics aspects of blood flow and dramatically improves vascular network outcomes. For instance, perfusing a medium through engineered tissues can prevent diffusion limitations and sustain cell viability, even in centimeter-scale constructs [111]. In one simple approach, a “wire molding” technique creates open microchannels in a cell-laden hydrogel, which are then perfused with media to nurture vessels. This has been shown to enhance cell survival and vessel formation deep inside the construct [112]. Such perfusion bioreactors effectively precondition the vascular network, promoting the formation of open lumens and endothelial junctions that resemble those in vivo.

In addition to macro-scale perfusion, microfluidic devices offer fine control over the microenvironment to build functional capillary networks in vitro. These devices contain microscale chambers and channels in which endothelial cells and support cells can be patterned in 3D matrices. By carefully controlling the flow rates and gradients, microfluidic platforms enable the self-assembly of capillaries that are already perfusable in vitro [113,114]. A landmark study combined tissue engineering with microfluidics to create a ~1 mm^3 fibrovascular tissue. Remarkably, it contained a living human capillary network that could be continuously perfused with a culture medium at physiological flow rates [113]. The engineered microvessels (15–50 μm diameter) were functional, with perfusion tests showing that they were largely impermeable to large solutes, indicating intact vessel walls [114]. This level of functionality is a direct result of providing convective flow and shear during network formation, something traditional static culture cannot achieve. Moreover, microfluidic vascularization chips can be designed to include “host” channels adjacent to the forming vessels, allowing researchers to observe and optimize how well engineered vessels anastomose with a perfused parent vessel analog. These organ-on-a-chip models serve as high-fidelity testbeds to refine in vitro networks so that they anastomose efficiently once implanted.

Another benefit of bioreactor culture is the ability to introduce mechanical stimuli beyond fluid flow. Cyclic stretch or pulsatile pressure can be applied to mimic the cardiac pulsation that vessels would experience in vivo, potentially encouraging the alignment and strengthening of endothelial cells and supporting mural cells [115]. Bioreactors can also maintain continuous perfusion up to the point of implantation. For example, a construct could be transferred from the bioreactor to the surgical site without a prolonged interruption in perfusion, minimizing ischemic time. While this is an area of active research, the early results are promising. Dynamically conditioned microvessel networks exhibit more in vivo-like morphology and gene expression, including an upregulation of junction proteins and basement membrane components [116]. In summary, perfused bioreactors and microfluidic culture devices represent powerful tools to engineer vascularized bone grafts. By recapitulating blood flow and mechanical cues, they address the maturity and stability issues of in vitro vessels, yielding prevascularized tissues with open, durable microvessels that are immediately perfusable and more likely to successfully inosculate with the host [111,112].

### 10.2. Novel Biomaterials and Scaffold Design

The design of scaffolds and biomaterials has a profound impact on vascular integration. Cutting-edge approaches in 3D bioprinting, electrospun nanofibers, and bioactive composite scaffolds are providing new ways to foster vessel formation in engineered bone. These strategies center on creating a physical environment that guides vascular network assembly and encourages host vessels to infiltrate and connect.

#### 10.2.1. 3D-Bioprinted Vascular Networks

Bioprinting enables the creation of complex, patient-specific scaffold geometries with an embedded vasculature. Researchers have used extrusion and inkjet bioprinters to deposit sacrificial filaments or hydrogel bioinks that define microchannel networks within bone scaffolds [117]. After printing, the sacrificial material can be removed to leave patent channels that act as artificial blood vessels. Endothelial cells can be lined within these channels, forming a printed capillary network throughout the construct. This approach allows for precise control over vessel architecture, diameter, and branching in the scaffold. For example, Kuss et al. printed a polycaprolactone/hydroxyapatite (PCL/HAp) bone scaffold and filled its pores with endothelial cell-laden hydrogel; the result was a prevascularized construct in which microvessels had already formed along the printed lattice prior to implantation [118]. In vivo, these predefined vessel networks facilitated more blood-filled connections to the host and improved perfusion vs. the non-printed controls [118]. Similarly, laser-assisted bioprinting has been used to directly pattern endothelial progenitor cells into desired locations within a scaffold, leading to rapid capillary assembly [119]. The advantage of 3D bioprinting is the ability to integrate channels or “vasculature” within large constructs in a designable way, overcoming diffusion limits by design. Ongoing improvements in bioprinting resolution and multi-material printing (for example, printing supportive perivascular cells alongside ECs) aim to produce more life-like microvascular trees. Bioprinted networks can also be made to align with the host vasculature geometry, for instance, by printing an inlet/outlet that can be surgically anastomosed to host vessels for immediate flow. In summary, bioprinting offers a versatile platform to embed pre-designed vascular networks into bone grafts, which can jump-start perfusion after implantation.

#### 10.2.2. Electrospun Nanofiber Scaffolds

Electrospinning produces fibrous mats that mimic the extracellular matrix (ECM) architecture, and these nanofiber scaffolds have shown great promise in promoting angiogenesis [120]. The high surface area of nanofibers and their ability to present adhesive cues make them excellent substrates for endothelial cell adhesion, migration, and capillary sprouting [121]. Moreover, electrospun scaffolds can be tuned in terms of fiber diameter, alignment, porosity, and stiffness—all factors that affect vascularization. Aligned nanofibers, for example, have been found to guide the directional growth of new microvessels, promoting organized neovascularization in vivo [122]. Fiber and pore size are also crucial. Sufficiently large pores within a fibrous scaffold (on the order of hundreds of microns) are needed to permit endothelial invasion and neovessel formation. Researchers recommend pore sizes of >300 µm for robust blood vessel infiltration in bone tissue scaffolds [123]. Electrospun materials can be functionalized to enhance vascularization. For instance, coating fibers with heparin or growth factor-binding domains can enable the retention and controlled release of VEGF, FGF-2, and other angiogenic factors [124]. One study showed that heparinized nanofibers significantly increased local angiogenic factor concentration and subsequent capillary density in the scaffold [124]. Additionally, electrospun scaffolds can be combined with other scaffold types (e.g., embedded in a 3D-printed frame or hydrogel) to create a composite that benefits from both a high-porosity framework and a bioactive fibrous matrix. The mechanical properties of nanofiber mats also play a role; surface stiffness influences endothelial cell spreading and sprouting [125]. By tailoring the fiber composition (e.g., incorporating biodegradable polymers that release neutral pH byproducts) and degradation rate, one can avoid acidic environments that might inhibit angiogenesis [126].

#### 10.2.3. Bioactive and Porous Scaffold Designs

Beyond fibers and printing, conventional scaffolds are being reinvented with angiogenesis in mind. Bioactive ceramics (such as calcium phosphate or bioactive glass) and composite polymer scaffolds are designed with optimized porosity to support both bone ingrowth and vascular penetration. It is now appreciated that a hierarchical pore structure is beneficial. Large interconnected pores (on the order of the diameter of small vessels) allow blood vessels to invade, while smaller micro-porosities increase the surface area for cell attachment and nutrient exchange [18]. New fabrication methods (freeze casting, gas foaming, etc.) can produce scaffolds with controllable macro-pore sizes and oriented channels. For example, Wang et al. created β-tricalcium phosphate scaffolds with arrays of longitudinal channels (~500 µm) and found markedly improved endothelial migration and angiogenesis in vivo [127]. Similarly, bionic scaffold structures—inspired by the natural cancellous bone architecture—have been 3D-printed or machined to direct vascular patterning. Incorporating hollow channels or vasculature-mimicking conduits in these scaffolds provides immediate pathways for blood flow post-implantation [128]. Another facet is adding bioactive molecules directly into the scaffold material, e.g., loading scaffolds with copper ions (which stabilize HIF-1α and promote angiogenesis) or with silicon ions (which can upregulate angiogenic factors) to create an inherently pro-vascularization material [129]. Growth factor delivery scaffolds, discussed more below, are also a form of bioactive design, using the scaffold as a vehicle to release VEGF, BMP-2, etc., at the defect site. Finally, advances in mechanically responsive biomaterials may address the compatibility issue. Materials that match bone’s stiffness or that can bear load initially and gradually transfer load to new tissue as they degrade. By minimizing stress shielding and providing the right mechanical cues (perhaps even via piezoelectric or magneto-responsive elements), these scaffolds may further promote the natural cascade of angiogenesis during bone healing [4,130]. In summary, novel scaffold designs are focusing on integrated functionality, not just serving as a tissue placeholder, but actively enhancing vascular ingrowth through pore architecture, bioactive coatings, and smart material choices. Such scaffolds, when combined with prevascularization techniques, greatly improve the likelihood of forming a well-perfused, integrated vascular network in the engineered bone.

### 10.3. Role of Growth Factors and Cell Sources

Biological factors, both the cells used to build vessels and the biochemical signals provided, are pivotal in engineering a stable and functional vasculature. Future strategies leverage optimal cell sources (like endothelial progenitors or stem-cell-derived endothelial cells) and spatiotemporally controlled delivery of growth factors (such as VEGF, FGF-2, BMP-2) to promote robust angiogenesis that is coupled with osteogenesis.

#### 10.3.1. Endothelial Progenitor Cells (EPCs)

EPCs isolated from peripheral blood or cord blood are an attractive cell source for prevascularization. These progenitors can differentiate into endothelial cells and have a high intrinsic capillary-forming potential [131]. In fact, EPCs often form vessels more readily than mature endothelial cells like HUVECs, in part because they proliferate robustly and secrete a suite of trophic factors. Notably, EPCs secrete osteogenic growth factors (e.g., BMPs and TGF-β) in addition to pro-angiogenic factors [131]. This means they can simultaneously stimulate blood vessel formation and bone regeneration, a valuable trait for coupling angiogenesis with osteogenesis. Co-cultures of EPCs with mesenchymal stem cells (MSCs) have demonstrated accelerated formation of capillary networks in vitro and improved vascularization in vivo, compared to using fully differentiated endothelial cells [98]. EPCs also present a lower risk of immunogenicity when obtained autologously (e.g., from the patient’s blood). The use of EPCs in 3D bioprinting has also been reported, such as, for instance, printing EPCs in defined patterns within a bone graft led to in situ capillary network formation [132].

#### 10.3.2. Induced Pluripotent Stem Cell-Derived ECs (iPSC-ECs)

Another frontier is using endothelial cells derived from induced pluripotent stem cells (iPSC-ECs). These cells offer an unlimited, patient-specific endothelial source. Recent studies have shown that iPSC-ECs can form vascular networks in vitro that anastomose with host vasculature after implantation [133]. For example, in a 2021 study, human iPSC-ECs were co-cultured with MSCs to generate prevascularized spheroids, which were then incorporated into a bone graft. Upon implantation in a cranial defect, the human iPSC-derived vessels connected with the mouse circulatory system within one week [133]. While the impact on bone volume in that particular study was modest, it demonstrated the feasibility of using iPSC-ECs to vascularize bone tissue. The advantage of an iPSC-derived endothelium is that it can be customized. For instance, cells can be engineered to express certain receptors or to secrete factors to enhance anastomosis. A challenge with iPSC-ECs is ensuring that they attain full maturity and stability; they may require additional maturation steps (such as flow conditioning or co-culture with pericytes) to behave like adult endothelial cells. Nonetheless, iPSC technology opens the door to patient-matched vascular cells for bone grafts, potentially eliminating immune rejection and providing a renewable source of cells for complex or large-scale tissue engineering.

#### 10.3.3. Growth Factors—VEGF, FGF-2, and BMP-2 (And Beyond)

The coordinated delivery of growth factors is key to guiding angiogenesis and osteogenesis in tandem. Vascular endothelial growth factor (VEGF) is arguably the most crucial angiogenic factor and has been a focal point of BTE strategies. VEGF initiates the sprouting of new vessels and also plays a central role in coupling blood vessel growth with bone formation [134,135]. Blocking VEGF signaling impairs both angiogenesis and osteogenesis in bone healing [135], while adding VEGF can enhance vascular invasion and bone repair [126]. However, a major lesson has been that the dose and release kinetics of VEGF must be carefully controlled. An initial burst of very high VEGF can lead to aberrant, leaky angiogenesis, whereas a lower, sustained release produces more mature and stable vasculature [126,136]. FGF-2 is another potent pro-angiogenic and pro-osteogenic factor that has been used to support vessel formation and maintenance. BMP-2 is widely known for its osteoinductive capacity, but also significantly influences vascularization by mobilizing endothelial progenitors and upregulating VEGF in osteogenic cells [137,138]. In tissue engineering, BMP-2 is often combined with an angiogenic factor for dual delivery. For instance, VEGF and BMP-2 dual delivery has been investigated in various models. One study encapsulated both in polymer microcarriers and found significantly greater blood vessel formation and bone regeneration compared to either factor alone [139]. The timing of delivery is also a frontier, e.g., delivering VEGF early and BMP-2 slightly delayed may mirror the natural sequence in fracture healing (blood vessels form and then bone mineralizes). In practice, scaffold-based delivery systems are being fine-tuned to release these factors in stages. By giving each signal at the right time, the aim is to create a pro-angiogenic and then pro-osteogenic microenvironment that yields a well-vascularized bone tissue. Additionally, using the right cell sources with these factors is key. MSCs in the construct can respond to BMP-2 by differentiating into bone, while responding to VEGF/FGF by upregulating blood vessel guidance cues. EPCs or iPSC-ECs will respond to VEGF/FGF by forming vessels, and PDGF by recruiting mural cells. Thus, the interplay of the chosen cells and delivered factors must be considered in tandem.

Overall, future strategies for prevascularized bone grafts are converging on a multifaceted approach: dynamic culture systems (bioreactors) to produce mature vessels in vitro, innovative scaffolds that present the right structural and mechanical environment, and smart use of cells and factors to drive angiogenesis alongside osteogenesis. By addressing the challenges of network maturity, integration, and cell survival from multiple angles, these approaches are bringing the field closer to the goal of clinically transplantable, large-scale bone constructs with a built-in blood supply. The progress of the last 20 years—from early co-culture studies to today’s bioprinted, growth-factor-loaded, perfused bone organoids—paints an optimistic picture that fully vascularized engineered bone grafts are on the horizon [2,140]. Various scaffold modifications, such as gradient porosity, micro-channels, or bioactive ceramic coatings, have demonstrated improved vascular infiltration. Meanwhile, Table 4 provides a concise overview of the in vivo animal models (rodent, rabbit, and larger species) described within this manuscript, emphasizing the defect site, species, experimental purpose, and references.

## 11. Commercialization Barriers and Emerging Off-the-Shelf Vascularized Bone Scaffolds

Despite extensive research into vascularized bone tissue engineering, there are still no fully commercialized off-the-shelf 3D scaffolds that integrate both vascular and bone components and are ready for clinical implantation [141]. Translating such complex constructs from bench to bedside has been impeded by several major hurdles, including scale-up and production scalability challenges, complex and protracted regulatory approval pathways, manufacturing and quality-control complexity, and high development costs [142]. The multi-component nature of a pre-vascularized bone graft (combining osteogenic scaffolds with vascular networks or cells) compounds these issues, making consistent mass production and regulatory compliance exceedingly difficult under current frameworks [141]. Consequently, no off-the-shelf vascularized bone graft has yet achieved full commercialization. Nevertheless, a few near-commercial strategies, for example, scaffold systems incorporating stem cells or pro-angiogenic factors, have shown promising results in late-stage development and early clinical studies, suggesting that viable vascularized bone graft solutions may be on the horizon and warrant continued attention.

## 12. Conclusions

Vascularization has emerged as a cornerstone of successful bone tissue engineering, as a functional blood supply is essential for nutrient delivery, waste removal, and overall tissue viability. Without pre-established vascular networks, even well-constructed bone grafts often fail due to central necrosis and poor integration, underscoring the need to incorporate vasculature into engineered bone constructs. Over the past decade, substantial progress has been made in overcoming this challenge. A diverse array of engineering approaches now enables the formation of perfusable vascular networks within bone graft substitutes. These include advanced fabrication techniques such as 3D bioprinting to directly pattern vascularized bone tissues, the use of electrospun nanofiber scaffolds that mimic the extracellular matrix and facilitate microvessel ingrowth, the optimization of scaffold architectures (e.g., porous or microchannel-containing biomaterials) to promote angiogenesis, the controlled delivery of angiogenic growth factors (like VEGF or BMP-2) to stimulate blood vessel formation, and cellular strategies like endothelial cell co-culture and in vitro or in vivo prevascularization of constructs. Importantly, extensive preclinical evidence supports the efficacy of these strategies. Vascularized bone constructs produced with these approaches consistently demonstrate improved outcomes in animal models, including greater tissue survival, more robust vascular infiltration, and enhanced bone regeneration compared to non-vascularized counterparts. For example, prevascularized implants created via 3D bioprinting have shown significantly increased blood vessel formation and bone healing in critical-sized defect models. Likewise, incorporating structural cues and nanoscale features in scaffolds accelerates host vessel in-growth, while a sustained release of growth factors (such as VEGF) from biomaterials promotes neovascularization and new bone tissue formation. Cell-based approaches further amplify these effects. Co-culturing mesenchymal stromal cells with endothelial cells or adding endothelial progenitors results in more extensive microvessel networks and concomitant osteogenesis. Collectively, these advancements highlight a marked improvement in our ability to recapitulate the vascularized bone microenvironment. Although challenges remain in achieving fully mature and long-lasting vessel networks and in translating these complex tissue constructs to clinical settings, the preclinical successes to date represent a significant step forward. By leveraging multi-faceted approaches to vascularize engineered bone, researchers have greatly enhanced graft viability and regenerative performance, thereby solidifying the critical role of vascularization in bone tissue engineering.

## Figures and Tables

**Figure 1 biomimetics-10-00412-f001:**
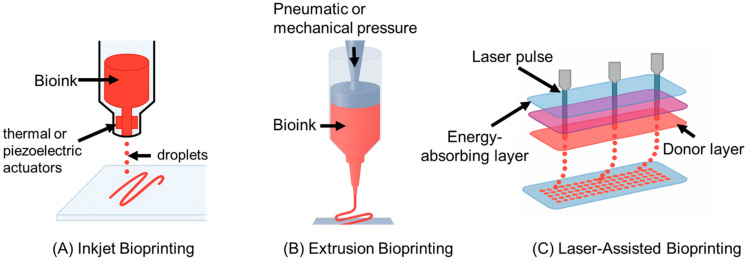
(**A**) Inkjet bioprinting uses thermal or piezoelectric actuators to deposit picoliter-sized droplets of low-viscosity bioinks in precise patterns. It enables high cell viability and resolution, making it suitable for spatial delivery of cells or growth factors, though limited by material viscosity and potential nozzle clogging. (**B**) Extrusion bioprinting extrudes continuous filaments of high-viscosity bioinks using pneumatic or mechanical pressure. It allows for printing robust, multi-material bone constructs and supports vascularization through techniques like coaxial extrusion. However, it has lower resolution and may impose shear stress on cells. (**C**) Laser-assisted bioprinting (LAB) employs a focused laser pulse to transfer cell-laden droplets from a donor surface without a nozzle. LAB offers the highest resolution and cell viability among the three methods and is well-suited for microscale cell patterning. It is less scalable for large constructs but is valuable for seeding fine vascular networks.

**Table 1 biomimetics-10-00412-t001:** 3D-bioprinting technologies.

Technology	Characteristics	Advantages	Disadvantages	Key References
Inkjet Bioprinting	Droplet-based, non-contact printing; requires low-viscosity bioinks; offers ~50–100 μm placement accuracy and moderate cell density.	Fast, low-cost, highly viable (>85%) cell deposition; supports multi-material patterning.	Limited to thin, low-viscosity inks; nozzle clogging risk and uneven droplets with dense or viscous bioinks.	[13,14]
Laser-Assisted Bioprinting	Nozzle-free laser pulses eject droplets; handles high-viscosity, high-cell-density inks; single-cell to tens of μm precision.	Highest spatial resolution and >95% viability; prints dense, complex patterns without clogging.	Expensive, complex setup; slower throughput as each droplet is shot individually; preparation of donor “ribbon” adds time.	[15,16]
Extrusion Bioprinting	Continuous filament extrusion via pneumatic/piston force; accepts wide viscosity range; builds layered, cm-scale scaffolds.	Most versatile: natural/synthetic bioinks, high cell loads, and large anatomical shapes; simple, widely available hardware.	Lower (~200–1000 μm) resolution; shear stress can drop viability to 40–90%; thicker strands reduce fine micro-architecture.	[17,18]

**Table 2 biomimetics-10-00412-t002:** Cell types used.

Cell Type	Main Function	Specific Role in Bone Engineering	Key References
Mesenchymal Stem Cells (MSCs)	Adult multipotent progenitors that secrete trophic cytokines and can differentiate into osteogenic, chondrogenic, and other mesenchymal lineages.	Primary osteoprogenitors; release VEGF/SDF-1 to recruit endothelial cells, thereby coupling osteogenesis and angiogenesis in co-culture systems.	[19,25]
Endothelial Cells (ECs)	Vascular lining cells that proliferate, migrate, and self-assemble into capillaries during angiogenesis.	Form perfusable microvascular networks within thick bone grafts and exchange VEGF/BMP signals with osteogenic cells to accelerate bone formation.	[20,26]
Osteoblasts	Matrix-producing bone cells that secrete type I collagen and initiate hydroxyapatite mineralization; mature into osteocytes.	Build mineralized ECM inside scaffolds, providing structural strength and promoting mechanical integration with host bone.	[21,27]
Pericytes	Perivascular mural cells that wrap capillaries, regulate flow, and stabilize nascent vessels.	Co-culture with ECs prevents vessel regression and sustains perfusion, indirectly supporting osteogenesis in engineered constructs.	[22,28]

**Table 3 biomimetics-10-00412-t003:** Growth factors.

Growth Factor	Main Action	Release Method	Key References
VEGF (Vascular Endothelial Growth Factor)	Principal angiogenic driver; also promotes coupled osteogenesis.	Sustained carriers (e.g., PLGA or hydrogel microspheres); often co-delivered with BMP-2 for synergistic bone–vessel ingrowth.	[29,30]
FGF-2 (Fibroblast Growth Factor 2)	Potent mitogen that boosts endothelial and fibroblast proliferation, enhancing vascularization and complementing VEGF.	Incorporated in ECM-binding hydrogels or nanoparticles for prolonged signaling during early repair.	[31,32]
PDGF-BB (Platelet-Derived Growth Factor)	Recruits pericytes/smooth muscle cells to mature and stabilize new vessels; modulates inflammation.	Gelatin or polymer slow-release systems; frequently paired with VEGF in sequential delivery profiles.	[35,36]
TGF-β1 (Transforming Growth Factor Beta 1)	Multifunctional factor: attracts mural cells, regulates ECM, and supports osteogenic or endochondral pathways.	Controlled microsphere or hydrogel release to avoid fibrosis; used early, with later factors guiding mineralization.	[33,37]
Angiopoietin-1 (Ang1)	Key vessel-maturation ligand acting with VEGF; strengthens and seals nascent capillaries.	Delivered with VEGF (simultaneous/sequential) via hydrogels or gene vectors for sustained local expression.	[34,38]
BMP-2 (Bone Morphogenetic Protein-2)	Powerful osteoinducer; indirectly boosts angiogenesis via VEGF up-regulation.	Collagen sponges or polymeric microspheres for controlled delivery; often co-administered with VEGF in bioprinted scaffolds.	[39,40]

**Table 4 biomimetics-10-00412-t004:** In vivo animal models for vascularized bone grafts.

Model (Body Site)	Animal Species	Purpose	Reference Number
Rat calvarial defect (critical size)	Rat	Bone regeneration assessment in cranial defect (non-load bearing)	[36]
Rat femoral segmental defect	Rat	Critical-size long bone defect healing under weight-bearing conditions	[45]
Rat arteriovenous (AV) loop model (groin)	Rat	Prefabrication of vascularized bone grafts (evaluate vascularization and bone formation)	[78]
Rabbit radial segmental defect	Rabbit	Bone regeneration evaluation in a medium-sized model (forelimb segmental defect)	[56]
Canine tibial segmental defect	Dog	Large defect repair and graft viability assessment in a large animal model	[90]
Sheep tibial segmental defect	Sheep	Bone regeneration in a large weight-bearing defect (clinically relevant model)	[85]

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
