# Peer review of "Preclinical Evaluation and Advancements in Vascularized Bone Tissue Engineering"

_biomimetics, 2025, doi:10.3390/biomimetics10070412_

Round 1

Reviewer 1 Report

Comments and Suggestions for Authors

Comments for manuscript titled ‘Preclinical Evaluation and Advancements in Vascularized Bone Tissue Engineering’

The manuscript ‘Preclinical Evaluation and Advancements in Vascularized Bone Tissue Engineering’ is a well-structured review with providing an overview of the major developments in the emerging field of vascularized bone tissue engineering. The authors have explained the current research gaps in the field very well and have provided examples whereever applicable. The quality of the review can be further improved to enhance by addressing/incorporating the following points:

General comments:

  1. Throughout the manuscript, please italicize terms like -in vitro, -in vivo,
  2. For all the tables (except Table 4) in the manuscript: There is too much information in the tables. For the ease of readers, please make it brief highlighting only key features and put in a bulleted form.

Graphical abstract:

  1. Including a graphical abstract would be helpful for the readers and will make the manuscript more impactful.

Abstract:
3. The abstract is too long. Please make it more concise. The journal guideline is 200 words maximum and a single paragraph, without headings but structured in the form: 1) Background: Place the question addressed in a broad context and highlight the purpose of the study; 2) Methods: Describe briefly the main methods or treatments applied. Include any relevant preregistration numbers, and species and strains of any animals used; 3) Results: Summarize the article's main findings; and 4) Conclusion: Indicate the main conclusions or interpretations.

Keywords:
4. The authors have not added any keywords. Please add three to ten pertinent keywords after the abstract. The keywords should be specific to the article, yet reasonably common within the subject discipline.

Section1: Introduction:

  1. The introduction is very well-structured with a good flow. At the end of the section, please include a brief paragraph emphasizing on how this review builds up on the existing knowledge in this field.

Section 2: 3D Bioprinting Approaches

  1. In this section, the authors are comparing three techniques for 3D Bioprinting (Inkjet, Extrusion, Laser-Assisted). Adding a schematic representation for each technique would be quite impactful.
  2. Section 2.2- Preclinical and In Vivo Successes in Vascularized Bone Bioprinting- Put the discussed case studies year-wise.
  3. The authors have mentioned highlights of these works, but individually also add 1) How a discussed study added to the existing knowledge in this field; and 2) Limitations of the study for each case. By doing this, the authors will critically review these works and help establish gaps that were filled by later studies or are yet to be bridged.
  4. Out of all the available techniques for fabrication of scaffolds, why have the authors dedicated a separate section for ‘Electrospun Nanofiber Scaffolds’. Please provide a rationale at the beginning of this section. The authors can highlight the fact that this technique consists of favorable conditions during synthesis process that helps to maintain the integrity and functionality of the sensitive molecules like growth factors that support vascularization.

  1. Authors can emphasize on the fact that, at the moment, there are no fully commercialized, off-the-shelf 3D scaffolds specifically designed for vascularized bone tissue engineering that include both vascular and bone components ready for implantation. Then briefly discuss the hurdles that the products are facing. It would be good to mention a few examples that near-commercial and worth-watching (Xylyx Bio (ECM-derived materials); Vasculogenic bioprinted scaffolds from Prellis Biologics; EpiBone (developing patient-specific bone grafts with vascularization strategies – clinical stage)

References:

11. Please cite more references from recent studies, literature reviews, clinical studies and emerging trends.

Author Response

Dear Editor and Esteemed Reviewers,

We sincerely thank you for the thorough evaluation of our manuscript, “Preclinical Evaluation and Advancements in Vascularized Bone Tissue Engineering.” Your insightful comments and constructive suggestions have been invaluable in refining the work. We have carefully addressed every point raised, making substantial revisions to improve clarity, concision, and scientific rigor.

Reviewer 1

Comments for manuscript titled ‘Preclinical Evaluation and Advancements in Vascularized Bone Tissue Engineering’

The manuscript ‘Preclinical Evaluation and Advancements in Vascularized Bone Tissue Engineering’ is a well-structured review with providing an overview of the major developments in the emerging field of vascularized bone tissue engineering. The authors have explained the current research gaps in the field very well and have provided examples whereever applicable. The quality of the review can be further improved to enhance by addressing/incorporating the following points:

Answer:

Thank you for the positive comments.

General comments:

  1. Throughout the manuscript, please italicize terms like -in vitro, -in vivo,

Answer:

As you pointed out, we have changed them to italics.

Comment:

  1. For all the tables (except Table 4) in the manuscript: There is too much information in the tables. For the ease of readers, please make it brief highlighting only key features and put in a bulleted form.

Answer:

Thank you for pointing this out. We have significantly simplified the wording within the cells of Tables 1, 2, and 3.

Graphical abstract:

  1. Including a graphical abstract would be helpful for the readers and will make the manuscript more impactful.

Answer:

That is an excellent suggestion. A graphical abstract would certainly attract readers even more; however, producing a figure that fully conveys the paper’s concept is difficult within the current revision period. Instead, we have streamlined the textual abstract and hope this will aid readers’ understanding.

Comment:

Abstract:
3. The abstract is too long. Please make it more concise. The journal guideline is 200 words maximum and a single paragraph, without headings but structured in the form: 1) Background: Place the question addressed in a broad context and highlight the purpose of the study; 2) Methods: Describe briefly the main methods or treatments applied. Include any relevant preregistration numbers, and species and strains of any animals used; 3) Results: Summarize the article's main findings; and 4) Conclusion: Indicate the main conclusions or interpretations.

Answer:

Thank you for your valuable comment. We have revised the Abstract to be more focused and concise, reducing it to within the 200-word limit.

Comment:

Keywords:
4. The authors have not added any keywords. Please add three to ten pertinent keywords after the abstract. The keywords should be specific to the article, yet reasonably common within the subject discipline.

Answer:

Thank you for your suggestion. We have listed ten key words accordingly.

Comment:

Section1: Introduction:

  1. The introduction is very well-structured with a good flow. At the end of the section, please include a brief paragraph emphasizing on how this review builds up on the existing knowledge in this field.

Answer:

Thank you for your valuable suggestion. As requested, we have added a brief paragraph at the end of the Introduction.

Section 2: 3D Bioprinting Approaches

  1. In this section, the authors are comparing three techniques for 3D Bioprinting (Inkjet, Extrusion, Laser-Assisted). Adding a schematic representation for each technique would be quite impactful.

Answer:

Thank you for the suggestion. Including schematics would certainly enhance clarity; however, producing new figures for each technology is not feasible within the current revision period. Instead, we have simplified the comparison table for the technologies to aid readers’ understanding.

  1. Section 2.2- Preclinical and In Vivo Successes in Vascularized Bone Bioprinting- Put the discussed case studies year-wise.

Answer:

As requested, I have arranged them in chronological order of publication year.

  1. The authors have mentioned highlights of these works, but individually also add 1) How a discussed study added to the existing knowledge in this field; and 2) Limitations of the study for each case. By doing this, the authors will critically review these works and help establish gaps that were filled by later studies or are yet to be bridged.

Answer:

Thank you for the excellent suggestion. As advised, we have now added statements detailing each study’s contribution to the existing body of knowledge in this field and its specific limitations.

  1. Out of all the available techniques for fabrication of scaffolds, why have the authors dedicated a separate section for ‘Electrospun Nanofiber Scaffolds’. Please provide a rationale at the beginning of this section. The authors can highlight the fact that this technique consists of favorable conditions during synthesis process that helps to maintain the integrity and functionality of the sensitive molecules like growth factors that support vascularization.

Answer:

Thank you for your insightful comment. We appreciate the opportunity to clarify the rationale behind dedicating a separate section to Electrospun Nanofiber Scaffolds. Electrospinning has been highlighted distinctly due to its unique advantages in fabricating scaffolds that closely mimic the extracellular matrix (ECM) of bone, featuring high surface area and tunable porosity. Specifically, electrospinning allows for favorable processing conditions that are conducive to preserving the integrity and functionality of sensitive bioactive molecules, such as growth factors, crucial for promoting vascularization. This attribute significantly enhances endothelial cell adhesion and proliferation, thus directly addressing the longstanding challenge of vascularizing engineered bone constructs. To explicitly address your comment, we have added this rationale at the beginning of the section on Electrospun Nanofiber Scaffolds in the revised manuscript.

  1. Authors can emphasize on the fact that, at the moment, there are no fully commercialized, off-the-shelf 3D scaffolds specifically designed for vascularized bone tissue engineering that include both vascular and bone components ready for implantation. Then briefly discuss the hurdles that the products are facing. It would be good to mention a few examples that near-commercial and worth-watching (Xylyx Bio (ECM-derived materials); Vasculogenic bioprinted scaffolds from Prellis Biologics; EpiBone (developing patient-specific bone grafts with vascularization strategies – clinical stage)

Answer:

Thank you for your extremely valuable suggestion. In accordance with your recommendation, we have drafted the paragraph and inserted it immediately before the Conclusion section.

References:

  1. Please cite more references from recent studies, literature reviews, clinical studies and emerging trends.

 Answer:

We have added two new citations, bringing the total number of key papers from the past five years to 26. For the foundational technologies, we have preferentially cited the earliest landmark publications whenever possible.

Reviewer 2

The review presents an interesting and relevant problem regarding vascularized bone tissue engineering. The manuscript is well-written and organized with relevant literature references, an adequate number of cited publications, and is free of visible errors. However, one editorial aspect, in my opinion, requires further clarification. Tables 1, 2, and 3 are difficult to read and, considering the amount of text, should cover the full page width. Additionally, highlighting subsequent rows would improve readability.

Answer:

As noted by the reviewers, the text within the cells of Tables 1, 2, and 3 was overly lengthy, so we have now shortened each entry to improve readability.

Reviewer 3

The authors had done a good job by providing a comprehensive review on the advancements in vascularized bone tissue engineering.  The manuscript began with the introduction of several techniques for fabricating vascularized bone grafts including inkjet bioprinting, extrusion and laser-assisted bioprinting. Several cell types used for bioprinting and growth factors were included. Moreover, in vivo animal model studies available from the literature relating to vascularized bone grafts were also addressed.  Finally, future strategies were provided and discussed. Based on these factors, this manuscript can be accepted for publication in this journal subjected to the following minor revision:

(1) The Abstract section is extremely long and bore. It looks like a brief Introduction of the manuscript. The authors should condense and shorten the Abstract considerably in the revised manuscript. Only key features relating to advancements in vascularized bone tissue engineering are addressed in the Abstract section. 

Answer:

Thank you for your valuable comment. We have revised the Abstract to be more focused and concise, reducing it to within the 200-word limit.

We trust that these revisions—together with the streamlined tables, and expanded discussion—significantly strengthen the manuscript and increase its value to the field. Should any further clarification be required, we would be pleased to respond promptly. Thank you again for your time and consideration, and we look forward to your favorable assessment of our resubmission.

Respectfully,

Toshiyuki Kawai

Reviewer 2 Report

Comments and Suggestions for Authors

The review presents an interesting and relevant problem regarding vascularized bone tissue engineering. The manuscript is well-written and organized with relevant literature references, an adequate number of cited publications, and is free of visible errors. However, one editorial aspect, in my opinion, requires further clarification. Tables 1, 2, and 3 are difficult to read and, considering the amount of text, should cover the full page width. Additionally, highlighting subsequent rows would improve readability.

Author Response

(The authors gave the same response as above.)

Reviewer 3 Report

Comments and Suggestions for Authors

The authors had done a good job by providing a comprehensive review on the advancements in vascularized bone tissue engineering.  The  manuscript began with the introduction of several techniques for fabricating vascularized bone grafts including inkjet bioprinting, extrusion and laser-assisted bioprinting. Several cell types used for bioprinting and growth factors were included. Moreover,  in vivo animal model studies available from the literature relating to vascularized bone grafts were also addressed.  Finally, future strategies were provided and discussed. Based on these factors, this manuscript can be accepted for publication in this journal subjected to the following minor revision:

(1) The Abstract section is extremely long and bore. It looks like a brief Introduction of the  manuscript. The authors should condense and shorten the Abstract considerably in the revised manuscript. Only key features relating to advancements in vascularized bone tissue engineering are addressed in the Abstract section. 

Author Response

(The authors gave the same response as above.)
